# Analysis of Factors Affecting Field Applicability and Long-Term Performance Analysis of LCP Woven Geotextile for Soft Ground Reinforcement

Yu Yan [1], Wangyu Hahm [2,*], Seunghyun Kim [3,*], Jiho Youk [3,*] and Hanyong Jeon [3,*]

[1] Industrial Science and Technology Research Institute, Inha University, Incheon 22212, Korea; dusdn@inha.ac.kr
[2] Advanced Textile R&D Department, Research Institute of Convergence Technology, Korea Institute of Industrial Technology, Ansan 15588, Korea
[3] Department of Chemical Engineering, Inha University, Incheon 22212, Korea
* Correspondence: wghahm@kitech.re.kr (W.H.); shk@inha.ac.kr (S.K.); youk@inha.ac.kr (J.Y.); hyjeon@inha.ac.kr (H.J.); Tel.: +82-31-8040-6251 (W.H.); +82-32-860-7493 (S.K.); +82-32-860-7498 (J.Y.); +82-32-872-1426 (H.J.)

**Abstract:** In recent years, natural disasters have been increasing worldwide due to rapid climate change, and the damage to ground structures is increasing due to the destruction of the ground. Damage to the ground structure can be reduced or eliminated by using LCP woven geotextiles as ground reinforcement. Therefore, in this study, the tensile properties, reduction factor affecting long-term performance, creep behavior, and fatigue properties of LCP woven geotextile were tested and analyzed. As a result, in the case of tensile properties, the maximum tensile strength of the LCP woven geotextile was 192.94 kN/m$^2$ in the MD direction, and it was generally constructed so that the load was transmitted. The total reduction factor is 1.86, which could be applied within 53.8% of the design strength when applied to the field. In addition, it was considered that the effect of the reduction factor for creep deformation on the long-term performance was dominant. Through the analysis of the creep behavior and fatigue characteristics, considering that the creep limit strain was 10%, if an earthquake occurred after 50 years of construction, it can be predicted that up to 90% of UTS would exhibit seismic performance. When LCP woven geotextile was applied as reinforcement, if the cyclic load due to fatigue failure was less than 478,000 times per year, it was considered that there was little possibility of the collapse of the ground structure.

**Keywords:** LCP woven geotextile; ground reinforcement; reduction factor; creep; fatigue

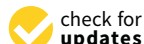



## 1. Introduction

Geotextile is one of geosynthetics used with sand, soil, gravel and water [1]. In the fields of civil engineering, construction, and environment, it is most widely used and widely applied in functions such as separation, reinforcement, filtration, and drainage [2,3]. Geotextiles are used for the purposes of crack prevention, ground structure protection, waste landfill, soft ground reinforcement, road construction, and retaining wall and slope protection [4,5]. Therefore, according to the characteristics of the ground reinforcement material, woven geotextiles used for ground reinforcement purposes need to have excellent physical properties such as strength, elongation, creep, earthquake resistance, and durability. In addition, due to the recent rapid climate change around the world, natural disasters such as heavy rains, tidal waves, and earthquakes are increasing, resulting in frequent ground destruction [6,7]. In consideration of this situation, in order to prevent destruction and deformation of ground structures and to ensure stability, we need woven geotextiles with better physical properties than the existing PP and PET woven geotextiles. Liquid crystal polymer (LCP) fiber is a high-performance, very durable, and mass-produced fiber

that creates anisotropy by melt spinning and promotes polymer orientation by suppressing chain entanglement with a less flexible, rigid-chain polymer. The first commercially available, melt-spun LCP fiber was introduced in 1990 under the brand name Vectran®, now manufactured by Kuraray, and it is also produced by companies such as Vectra and Dupont's Zenite. In addition, although LCP is expensive, it is attracting attention for its mass productivity, excellent physical properties, and plasticity [8,9].

So far, many studies have been conducted. In a paper on improving the engineering properties and the effect of this on long-term performance and on the field applicability of geotextiles, Chae et al. analyzed and considered the effect on the ground of geotextiles when the seaming strength of the geotextiles was improved through a study of geotextile seaming tensile strength and ground stress increment analysis. It has been reported that the improvement of the tensile strength of geotextiles has a great influence on the rate of increase in the bearing capacity of ground [10]. Yan et al. analyzed engineering properties, reduction factors affecting reinforcement, and long-term performance through a study on the long-term performance of seaming strength and factor analysis on the seaming area of a PET woven geotextile for reinforcing soft ground. It was reported that it was possible to improve the stability of soft ground and ground by improving the seaming strength of geotextiles through effective adhesive application [11]. Kim et al. verified the horizontal drainage capacity of geotextiles with separation and reinforcement performance through an experimental study to develop a geotextile for separation and reinforcement with horizontal wicking drainage characteristics. It was confirmed that the function of discharging excess pore water in the horizontal direction could be sufficiently exhibited [12]. Jeon et al. conducted a study on the resistance to the application environment of geotextiles. As a result of analyzing and examining the mechanical properties, ultraviolet resistance, and chemical stability of a composite geotextile, it was reported that the more regenerated polyester geotextiles were used, the better the ultraviolet stability, and the difference in factor strength varied depending on chemical conditions [13]. Han et al. conducted a study on the long-term performance of a PET geotextile using a chemical resistance test and an accelerated life analysis. According to the results of the chemical resistance test, the strength decreased as the temperature increased. The accelerated life analysis could predict the strength retention times of the geotextile [14]. Jeon et al. reported that reinforcing polyester woven geotextiles had a sufficient reinforcing performance similar to geogrids through long-term performance evaluation studies of reinforcing composite geotextiles [15]. Jeon et al. reported the results of the limiting creep strain using Sherby–Dorm plots proposed through a study on the limiting creep strain analysis of geogrids. When applied to design, it was reported that accurate application of materials was possible, and economic feasibility could be secured [16]. Ahn et al. reported that an LCP woven geotextile could be manufactured by improving and supplementing the preparation process and solving yarn-weaving problems by analyzing and considering the problems arising from securing the strength and expression of the low elongation of the LCP woven geotextile through research on the long-term performance prediction model analysis of the LCP woven geotextile [17].

After analyzing the preceding studies above, we found that theymainly showed the results of improved seaming tensile strength, reduction factor of long-term performance, drainage performance, and creep behavior of PET geogrids, due to improving the weaving process of the LCP woven geotextile. There are few studies on the tensile properties, reduction factors on field applicability, creep behavior, and fatigue properties of LCP woven geotextile. In consideration of these facts, this study confirmed the effects of the tensile properties and reduction factors on the field applicability of an LCP woven geotextile, and analyzed and examined the long-term durability due to creep behavior and fatigue characteristics.

## 2. Experimental

### 2.1. Preparation of Geotextiles

Polyester and polypropylene fibers are used as geotextile materials that are most commonly applied to the reinforcement of soft ground in Korea, but LCP fibers with better performance in areas such as tensile properties, low permeability, thermal properties, dimensional stability, creep, earthquake, durability, and various resistances (e.g., radiation, chemical, abrasion, impact, UV, cut) were used [18].

The sample used in this study was an LCP woven geotextile that was manufactured as a twill structure using 1000 denier Vectran® (Kuraray Co. Ltd., Tokyo, Japan) as a yarn for warp and weft. The properties of Vectran® yarn suggested by Kuraray are as follows: density of 1.4 g/cm$^3$, strength of 1.1 GPa, and elongation of 3.8%. Table 1 shows the specifications of the geotextile, and Figure 1 shows the fabric design diagram and a photograph of the geotextile.

**Table 1.** Specifications of LCP woven geotextile.

| LCP Woven Geotextile | Yarn | Weave | Yarn (Denier) | Density (g/cm$^3$) | Strength (GPa) | Elongation |
|---|---|---|---|---|---|---|
| Warp Weft | Vectran® | Twill | 1000 | 1.4 | 1.1 | 3.8% |

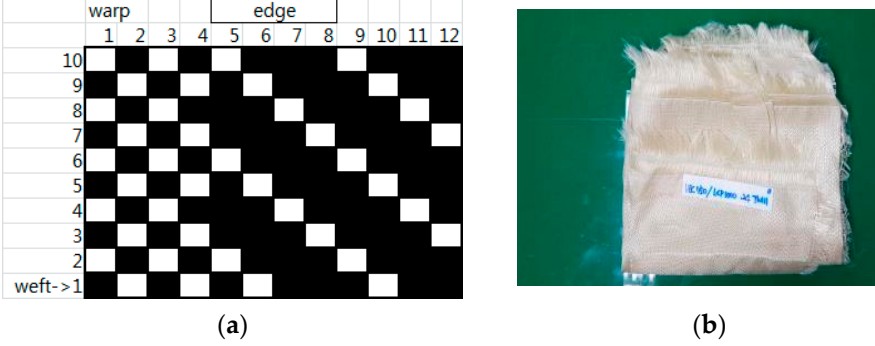

(**a**)        (**b**)

**Figure 1.** Fabric design diagram and photograph of geotextile. (**a**) Fabric design diagram; (**b**) LCP woven geotextile.

### 2.2. Engineering Performance Evaluation

(1)  Tensile properties

According to ISO 10319:2015 (Geosynthetics—Wide-width tensile test), we set the distance between the clamps of the CRE type tensile tester to 10 ± 0.3 cm. After mounting a specimen with a width of 20 cm, the test was conducted at a rate of 10 ± 3%/min until the specimen was broken to measure tensile stress, strain, and elastic modulus.

(2)  Creep deformation

The creep deformation test generally used requires a minimum of $10^4$ h of test time, representing a very time-consuming process. Therefore, in this paper, the test was conducted with a stepped isothermal method (SIM) based on the time–temperature superposition (TTS) principle with a short test period. The test method was in accordance with ASTM D6992-16 (Standard Test Method for Accelerated Tensile Creep and Creep-Rupture of Geosynthetic Materials Based on Time-Temperature Superposition Using the Stepped Isothermal Method). The distance between the webbing clamps was set to 10 ± 0.3 cm, and a 20 cm wide specimen was maintained in a standard condition at 20 ± 1 °C, and a prestress equivalent to 1% of the tensile strength was applied to the specimen before starting the test. In the case of the geotextile, the specimen was exposed to a five-stage temperature increase from 20 °C to 76 °C in steps of 14 °C, and data were recorded every

2 s at the beginning of the curve, and then every 60 s. Table 2 shows the experimental conditions, and Figure 2 shows the creep tester.

**Table 2.** Test design for creep deformation.

| Test Item | Isothermal Duration | Temperature (°C) | Loading Level (%) |
|---|---|---|---|
| Stepped isothermal method (SIM) | 10,000 s | 20, 34, 48, 62, 76 | 50~82 |

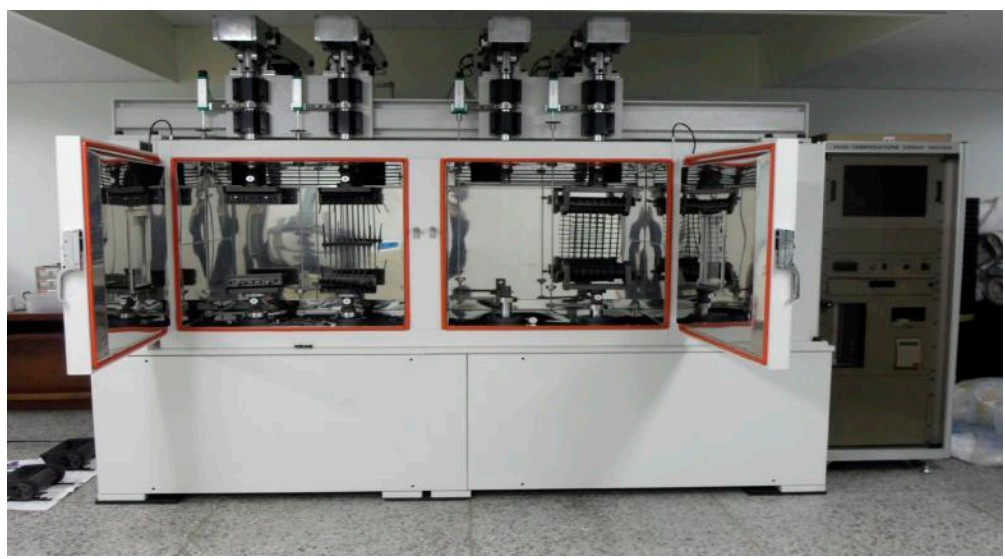

(**a**)

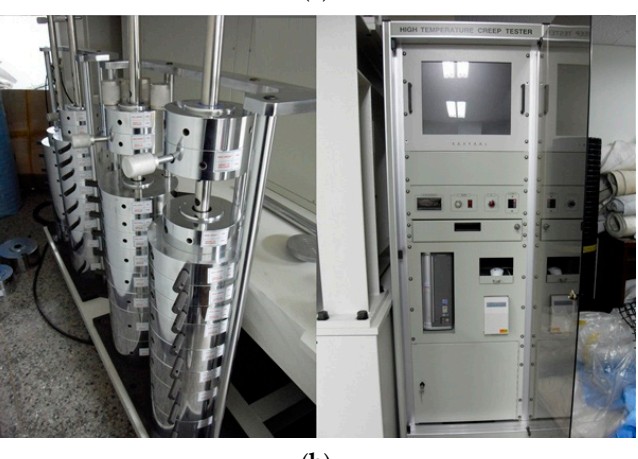

(**b**)

**Figure 2.** Photographs of creep test apparatus for SIM. (**a**) Creep test apparatus; (**b**) Loading system (**left**) and control system (**right**).

(3) UV resistance test

According to ASTM D4355-14 (Standard Test Method for Deterioration of Geotextiles by Exposure to Light, Moisture, and Heat in a Xenon Arc-Type Apparatus), exposure was performed for 20 days in a cycle of constant humidity and a certain time in a constant temperature and light exposure environment, and then tensile strength retention was measured and analyzed.

(4) Chemical degradation test

According to ASTM D5322-17 (Standard Practice for Laboratory Immersion Procedures for Evaluating the Chemical Resistance of Geosynthetics to Liquids), immersion was

performed at temperatures of 23 °C and 50 °C at pH 5, 11, and 13 for 60 days, and then tensile strength retention was measured and analyzed.

(5) Biological degradation test

According to BS EN 12225:2000 (Geotextiles and geotextile-related products—Method for determining the microbiological resistance by a soil burial test), short-term acceleration tests were performed in exposure and construction environments by microorganisms or bacteria in the soil, and then tensile strength retention was measured and analyzed.

(6) Evaluation of fatigue characteristics of the LCP woven geotextile

When a geotextile is applied to ground reinforcement, it has a function of preventing cracks generated by external force loads. If an external load is regularly added to the ground structure or an unexpected load is applied, it causes the ground structure to collapse due to fatigue destruction of the geotextile. In order to review this, a fatigue destruction test of geotextile was performed according to ASTM D7791-12 (Standard Test Method for Uniaxial Fatigue Properties of Plastics).

## 3. Results and Discussion

### 3.1. Tensile Properties of the LCP Woven Geotextile

In this study, an experiment was conducted to confirm the tensile characteristics of an LCP woven geotextile, and this study was conducted with the LCP woven geotextile having the tensile characteristics. Table 3 shows the tensile characteristics of warp and weft yarns of the LCP woven geotextile.

**Table 3.** Tensile characteristics of LCP woven geotextile in weft and warp directions.

| LCP Woven Geotextile | Tensile Strength | | | Strain | | Modulus |
|---|---|---|---|---|---|---|
| | kN/m$^2$ | % of UTS | g/d | % | Difference | kN/m$^2$ |
| MD | 192.94 | 100 | 25 | 2.8 | 100% | 7827.89 |
| CMD | 157.50 | 81.6 | 17 | 3.9 | 139% | 4423.99 |

The tensile properties of the LCP woven geotextile are shown in Table 3 through the test. As shown in the table, the tensile strength of the LCP woven geotextile in the machine direction (MD) was 192.94 kN/m$^2$ with the ultimate tensile strength (UTS), and the tensile strength of the yarn was 25 g/d, the strain was 2.8%, and the elastic modulus was 7827.89 kN/m$^2$. In the cross-machine direction (CMD), the tensile strength of the LCP woven geotextile was 157.50 kN/m$^2$, which was 81.6% of UTS, and the tensile strength of the yarn was 17 g/d and the strain was 3.9%, which was 39% higher than in the MD, and the elastic modulus was 4423.99 kN/m$^2$. Therefore, since the MD was 18.4% stronger than the CMD, it was in the direction of receiving force, and was generally constructed so that the load was transmitted to the MD. In addition, due to the nature of the weaving process of the geotextile, the tensile strength of the machine direction (MD) was high because the fiber was stretched during winding of the geotextile, and the strength increased. The test was commissioned by a testing institution, and the test results received only numerical values through the report. Only tensile strength, strain, and modulus were displayed in the report, and the graph was not provided, so it could not be displayed.

### 3.2. Reduction Factor of the LCP Woven Geotextile

(1) Reduction factor

When applying a woven geotextile to soft ground, and considering the load induced or applied within the ground, the maximum tensile strength value obtained through the index test is not applied as it is, but with a correction factor that considers the reduction factor of the tensile strength in the construction conditions. At this time, the correction factor is

called a reduction factor, and the type of reduction factor to be considered is selected and determined by the application field. It is calculated by applying the corresponding reduction factor in the following equation, which determines the long-term permissible strength.

For reference, the reduction factor was obtained from Equation (3), which determined the reduction factor in each case in Equation (2):

$$T_{allow} = T_{ult} \left[ \frac{1}{\prod RF} \right] \tag{1}$$

$$= T_{ult} \left[ \frac{1}{RF_{ID} \times RF_{CR} \times RF_{CD} \times RF_{BD} \times RF_{JC} \times \cdots} \right] \tag{2}$$

$$RF = \frac{Design\ strength}{Actual\ strength(measured\ value\ by\ test)} \tag{3}$$

where $T_{ult}$ = ultimate tensile strength; $T_{allow}$ = allowable tensile strength; $RF_{ID}$ = reduction factor for installation damage; $RF_{CR}$ = reduction factor for creep deformation; $RF_{CD}$ = reduction factor for chemical degradation; $RF_{BD}$ = reduction factor for biological degradation; and $RF_{JC}$ = reduction factor for joints.

Instead of the term of the reduction factor considered in Equation (1), the reduction factor that lowers the long-term allowable tensile strength, which is the long-term performance of woven geotextiles, should be considered. The equations applied in this study were as follows:

$$T_{allow} = T_{ult} \left[ \frac{1}{\prod RF} \right] \tag{4}$$

$$= T_{ult} \left[ \frac{1}{RF_{CR} \times RF_{UV} \times RF_{CD} \times RF_{BD}} \right] \tag{5}$$

where $T_{ult}$ = ultimate strength; $T_{allow}$ = allowable strength; $RF_{CR}$ = reduction factor for creep deformation; $RF_{UV}$ = reduction factor for UV resistance; $RF_{CD}$ = reduction factor for chemical degradation; and $RF_{BD}$ = reduction factor for biological degradation.

(2)  Creep deformation

A short-term accelerated creep test by the stepped isotherm method (SIM) was performed, and showed a strength retention rate of 62.9% with a tensile strength of 121.35 $kN/m^2$ within 10% of the creep limit deformation, and the reduction factor for creep deformation was 1.59, as shown in Table 4. The decrease in tensile strength due to creep was 37.1%, and the strength applicable to the design was 62.9% of the tensile strength, which seemed to have a significant effect on the tensile strength. The reduction factor for creep deformation was obtained from Equation (3), and the reduction coefficient of 1.59 meant that the LCP woven geotextile may be applicable within 62.9% of the design strength when applied in the field.

**Table 4.** Strength retention and reduction factor for creep deformation of LCP woven geotextile.

| LCP Woven Geotextile | Creep | |
|---|---|---|
| | **Before** | **After** |
| Strength retention ($kN/m^2$) | 192.94 | 121.35 (62.9%) |
| $RF_{CR}$ | 1.59 | |

(3)  UV resistance

Short-term accelerated tests were performed in a UV-exposed construction environment, and the strength retention rate and reduction factor for UV resistance were measured and analyzed through strength comparison before and after use. The results are shown in Table 5; the strength retention rate was 92.6%, and the reduction factor was 1.08. This was due to the reduction of the tensile strength due to the exposure to UV rays, and the

effect on the tensile strength from exposure to sunlight in the construction environment was thought to be small but not significant. Here, the reduction factor of 1.08 meant that the LCP woven geotextile may be applicable within the range of 92.6% of the design strength when applied in the field.

**Table 5.** Strength retention and reduction factor for UV resistance of LCP woven geotextile.

| LCP Woven Geotextile | UV Resistance | |
| --- | --- | --- |
| | Before | After |
| Strength retention (kN/m$^2$) | 192.94 | 178.65 (92.6%) |
| RF$_{UV}$ | 1.08 | |

(4)   Chemical degradation

Short-term accelerated tests were performed under exposure to soil pH (acid and alkalinity), temperature, and time, and the strength retention rate and reduction factor for chemical degradation were measured and analyzed through a strength comparison before and after use. The results are shown in Table 6; the strength retention rate was 98.0% in acid and 93.5% in base. The reduction factor was 1.02 in acid and 1.07 in base. This was thought to be due to the fact that LCP is relatively stable to most acids, alkalis, and solvents, as the decrease in tensile strength occurred due to exposure to acids and alkalis, and the effect was insignificant or not very effective. Here, the reduction coefficients of 1.02 and 1.07 meant that the LCP woven geotextile may be applicable within a range of 98.0% and 93.5% of the design strength when applied in the field.

**Table 6.** Strength retention and reduction factor for chemical degradation of LCP woven geotextile.

| LCP Woven Geotextile | Chemical Degradation | | | |
| --- | --- | --- | --- | --- |
| | Acid | | Base | |
| | Before | After | Before | After |
| Strength retention (kN/m$^2$) | 192.94 | 189.16 (98.0%) | 192.94 | 180.32 (93.5%) |
| RF$_{CD}$ | 1.02 | | 1.07 | |

(5)   Biological degradation

A short-term accelerated test was performed under exposure to microorganisms or bacteria in the soil, and the strength retention rate and reduction factor for biological degradation were measured and analyzed through a strength comparison before and after use. The results are shown in Table 7; the strength retention rate was 94.3%, and the reduction factor was 1.06. This was due to the decrease in tensile strength due to the exposure to microorganisms or bacteria, and it was thought that the effect on the tensile strength was not significant when exposed to microorganisms or bacteria in the soil. Here, the reduction factor of 1.06 meant that the LCP woven geotextile may be applicable within the range of 94.3% of the design strength when applied in the field.

**Table 7.** Strength retention and reduction factor for biological degradation of LCP woven geotextile.

| LCP Woven Geotextile | Biological Degradation | |
| --- | --- | --- |
| | Before | After |
| Strength retention (kN/m$^2$) | 192.94 | 182.02 (94.3%) |
| RF$_{BD}$ | 1.06 | |

(6)   Total reduction factor

In order to obtain the long-term performance of the LCP woven geotextile, the total reduction factor (ΠRF) was determined by applying the reduction factor for creep deformation, UV resistance, and chemical and biological degradation from Equation (5), which determined the long-term permissible strength.

Tables 8 and 9 show the total reduction factor and long-term allowable tensile strength. In Table 8, the reduction factors for creep deformation, UV resistance, and chemical and biological are given, and the total reduction factor was 1.86. Table 9 shows the long-term allowable tensile strength was 103.73 kN/m$^2$ in the MD and 84.68 kN/m$^2$ in the CMD. Here, the total reduction factor of 1.86 indirectly suggested that long-term allowable tensile strength stability is guaranteed within the range of 53.8% of UTS in the case of field application of the LCP woven geotextile.

**Table 8.** Total reduction factor of LCP woven geotextile's long-term performance model.

| Reduction Factor | $RF_{CR}$ | $RF_{UV}$ | $RF_{CD}$ | $RF_{BD}$ | ΠRF |
|---|---|---|---|---|---|
| | 1.59 | 1.08 | 1.02 | 1.06 | 1.86 |

**Table 9.** Allowable tensile strength of LCP woven geotextile's long-term performance model.

| Allowable Tensile Strength (kN/m$^2$) | ΠRF (Before Application) | | ΠRF (After Application) | |
|---|---|---|---|---|
| | MD | CMD | MD | CMD |
| | 192.94 | 157.50 | 103.73 (53.8%) | 84.68 (53.8%) |

*3.3. Analysis of the Creep Behavior of the LCP Woven Geotextile*

Creep behavior represents a deformation process with time at a stress lower than the strength of the material [19,20]. Typical tensile creep behavior is described in Figure 3 for creep strain (solid line) and creep strain (dotted line). Creep behavior can be divided into three stages: first, second, and third creep. In the first (or transient) stage, it is called transition creep; the initial deformation occurs and the strain increases nonlinearly as the deformation increases. In the second (or steady state) stage, the displacement increases linearly with time, resulting in almost constant strain. In the third stage, the deformation increases rapidly, leading to creep rupture, and in this stage, the creep strain increases rapidly.

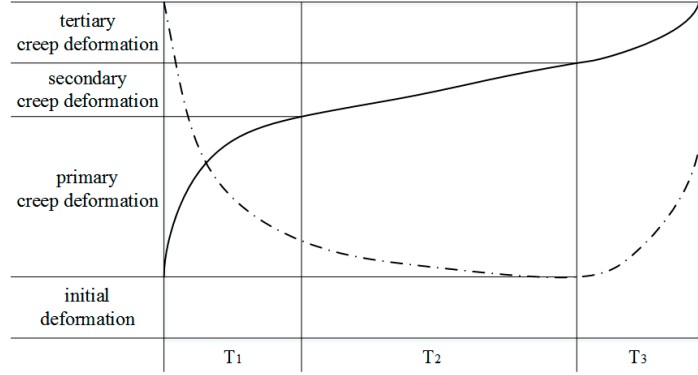

**Figure 3.** Typical creep strain and strain rate behavior with time. The solid line represents the creep strain, and the dashed line the creep strain rate [21].

In the case of the LCP woven geotextile, if an earthquake occurs during actual use, an earthquake load is added, and long-term performance is affected. Creep behavior caused by the addition of seismic loads was measured and analyzed according to the SIM. In the creep characteristics evaluation, creep characteristics simulating an earthquake situation

were evaluated by adding a load that considered earthquake conditions for 1 min at a specific point in time. The creep test was conducted at temperatures from 20 °C to 76 °C every 14 °C and with a load of 40–60% of UTS. After the creep test, the earthquake situation was simulated by additional testing for 1 min up to 80–100% of UTS after 1.5 h at 20 °C and 40–60% of UTS. The results are shown in Tables 10–12.

**Table 10.** Creep strain of LCP woven geotextile during creep test by 40% of UTS at initial and critical times.

| Creep Strain (%) | | Time (Hours) | |
|---|---|---|---|
| | | 0.01 | 438,000 |
| | 100% | 5.01 | 10.35 |
| Seismic Load | 90% | 5.00 | 8.75 |
| | 80% | 5.02 | 8.01 |

**Table 11.** Creep strain of LCP woven geotextile during creep test by 50% of UTS at initial and critical times.

| Creep Strain (%) | | Time (Hours) | |
|---|---|---|---|
| | | 0.01 | 438,000 |
| | 100% | 6.00 | 9.91 |
| Seismic Load | 90% | 6.03 | 8.80 |
| | 80% | 6.01 | 8.32 |

**Table 12.** Creep strain of LCP woven geotextile during creep test by 60% of UTS at initial and critical times.

| Creep Strain (%) | | Time (Hours) | |
|---|---|---|---|
| | | 0.01 | 438,000 |
| | 100% | 7.25 | 10.10 |
| Seismic Load | 90% | 7.26 | 9.71 |
| | 80% | 7.26 | 9.41 |

Tables 10–12 show the initial creep strain and the maximum creep strain by adding 80–100% of UTS for one minute at creep loads of 40–60% of UTS. As shown in the table, the initial creep strain was shown at the initial time (0.01 h), and the maximum creep strain occurred at the time when the seismic load was added (438,000 h). Thus, the creep strain increased as the creep load increased and the added seismic load increased. In addition, the initial creep strain was constant when the creep load was the same, and increased as the creep load increased. Here, the creep strain at 40% of UTS and 100% seismic load was 10.35%, which was larger than in the case of a 100% seismic load at 50% of UTS and 60% of UTS. However, the reason was considered to be that the LCP woven geotextile was partially nonuniform or damaged, resulting in a large creep strain.

The behavior of the LCP woven geotextile while assuming an earthquake occurrence 50 years after construction was predicted. Considering that the creep limit strain was 10%, it was predicted that the seismic load would exhibit a seismic resistance up to 90% of UTS of the LCP woven geotextile, assuming an earthquake 50 years after construction.

*3.4. Fatigue Characteristics of the LCP Woven Geotextile*

The fatigue characteristics of the LCP woven geotextile were analyzed by using the ratio of the crack growth rate and the number of repeated loads. The crack growth rate and expected fatigue life expectancy of reinforcement such as geotextiles were calculated using Equations (6) and (7). In general, the fatigue life of the LCP woven geotextile was measured and evaluated by citing K = 10 N/mm$^{1.5}$, A = 1.0 × 10$^{-8}$, n = 4.3, and design traffic

number = 100,000 times/year as values corresponding to woven geotextiles for reinforcement [22].

The crack growth rate and the fatigue life can be expressed as follows:

$$\frac{dc}{dN} = A \cdot K^n \tag{6}$$

$$\text{Lifetime} = L / \frac{dc}{dN} \tag{7}$$

where $\frac{dc}{dN}$ = crack growth rate for cyclic load; $K$ = stress concentrator; $A$, $n$ = experimental constant; and $L$ = thickness of the reinforcing layer.

The fatigue experiment using the LCP woven geotextile was conducted at a fatigue frequency of 5 Hz by applying a partial knitting load using a test sample with a width of 25 mm and a length of 300 mm at room temperature. In addition, the maximum tensile load was 50% of UTS, the minimum tensile load was 100 N/25 mm, and the tensile strength retention rate was calculated from the number of repetitions of fatigue loads at 50,000 times, 100,000 times, 400,000 times, and 600,000 times. The results were calculated by calculating the number of repetitions at the tensile strength retention rate of 90% through linear regression analysis between the number of repetitions and the tensile strength retention rate; the fatigue lifetime is shown in Table 13. When the LCP woven geotextile was used as a reinforcing material, the fatigue lifetime was calculated by the number of repetitions at a tensile strength retention rate of 90%. In the case of the LCP woven geotextile, this meant that it could withstand 478,000 cycles per year by cyclic loading. It was considered that when the LCP woven geotextile was applied as a reinforcing material, there was almost no possibility of collapse of the ground structure when the repeated loads due to fatigue failure were less than 478,000 times per year.

**Table 13.** Lifetime of LCP woven geotextile.

| Lifetime (Cycles/Year) | $4.78 \times 10^5$ |
|---|---|

## 4. Conclusions

When applying an LCP woven geotextile to reinforce the ground, it is necessary to prevent destruction and deformation of the ground structure and secure stability. Therefore, the tensile properties, creep deformation and reduction factors, creep behavior, and fatigue characteristics of an LCP woven geotextile were tested and analyzed.

As a result of testing the tensile properties of the LCP woven geotextile, the maximum tensile strength was 192.9 kN/m$^2$ in the MD. Therefore, it is constructed so that the MD is in the direction of receiving force, and generally the load is transmitted. In addition, due to the characteristics of the weaving process of the geotextile, the tensile strength of the machine direction (MD) was large, and it was believed that the strength increased due to elongation of the fibers in the process of winding the geotextile. In the case of the reduction factor, the total reduction factor was 1.86, and tensile strength retention rate was 53.8%. The geotextile can be applied within a range of 53.8% of the design strength when applied in the field. Here, the reduction factor for creep deformation was 1.59, which was considered to have a very large effect on the design strength. It was considered that the effect of the reduction factor for creep deformation on the tensile strength was dominant. In the case of creep behavior, from the results of analyzing the behavior of the LCP woven geotextile while assuming an earthquake 50 years after construction, if it was thought that the creep limit strain was 10% if an earthquake occurred during actual use, it could be predicted that seismic resistance would be expressed up to 90% of UTS 50 years after construction. In the case of fatigue characteristics, when the LCP woven geotextile was used as a reinforcing material, the results for fatigue lifetime were calculated by repetition at a tensile strength retention rate of 90%; when the LCP woven geotextile was applied as a reinforcing material,

there was almost no possibility of collapse of the ground structure when the repeated loads due to fatigue failure were less than 478,000 times per year.

Finally, through this study, the tensile characteristics of the LCP woven geotextile were confirmed, and it was judged that the reduction factor for creep deformation had a very large effect on long-term performance. In addition, after predicting seismic resistance in earthquake situations and fatigue life due to repetitive loads, when the LCP woven geotextile is applied in the field, it was believed that design conditions should consider the field more.

**Author Contributions:** Conceptualization, W.H., S.K., J.Y. and H.J.; methodology, H.J.; validation, Y.Y. and W.H.; formal analysis, Y.Y. and S.K.; investigation, Y.Y. and W.H.; data curation, J.Y.; writing—original draft preparation, Y.Y.; writing—review and editing, W.H., S.K., J.Y. and H.J.; supervision, W.H., J.Y. and H.J.; project administration, H.J.; funding acquisition, H.J. All authors have read and agreed to the published version of the manuscript.

**Funding:** This work was supported by Industrial Strategic Technology Development Program (Grant No. 20010999) funded by the Ministry of Trade, Industry & Energy (MOTIE, Korea).

**Data Availability Statement:** Not applicable.

**Conflicts of Interest:** The authors declare no conflict of interest.

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
