# Peer review of "Analysis of Factors Affecting Field Applicability and Long-Term Performance Analysis of LCP Woven Geotextile for Soft Ground Reinforcement"

_applsci, doi:10.3390/app12031345_

Round 1
Reviewer 1 Report
- Try to improve the summary. Eliminate obvious and redundant information. Provide more details based on the results obtained.
- The literature review should be expanded. Avoid collective citation. Try to accurately describe each publication.
- Enter the mechanical properties given by the manufacturer, Kuraray Co. Ltd., Japan.
- Specify exactly what the sample for specific tests looked like and how it was attached.
- There should be no theoretical description of the model in the methodology of the experiment. This excerpt should be placed in advance.
- In Table 3, the units are incorrectly given. Correct it. In the rest of the text, too.
- In the case of fatigue tests, the variability of the load must be accurately reported.
- Conclusions are to be based on the results. Describe the results exactly so that what you describe in the villages is consistent.
Reviewer 2 Report
applsci-1558771 – Review
Analysis of factors affecting field applicability and long-term performance analysis of LCP woven geotextile for soft ground reinforcement
General comment:
The paper presents the test results and analysis of tensile and fatigue properties as well as creep behavior of Liquid Crystal Polymers (LCP) woven geotextile to assess its long-term performance for soft ground reinforcement. In order to determine the creep deformation of LCP woven geotextile the method for Accelerated Tensile Creep and Creep-Rupture of Geosynthetic Materials Based on Time-Temperature Superposition Using the Stepped Isothermal Method (SIM) according to ASTM D6992-16 was used. In laboratory tests the tensile strengths, strains and module of LCP woven geotextile in machine direction (MD) and in cross machine direction (CMD) were determined. The reduction factor for creep deformation was determined. The fatigue life due to repeated loads and earthquake resistance were evaluated.
In my opinion, the paper text should be supplemented with additional information and short comments on:
- creep test methods other than the SIM method
- the authors assumed that other reduction factors are equal to 1.0, hence total reduction factor equals reduction factor for creep deformation. In practice, the other reduction factors are not equal to 1.0. This requires additional comment (see for example EBGEO - Recommendations for Design and Analysis of Earth Structures using Geosynthetic Rainforcements – DGGT, Ernst & Sohn, 2011
- in addition to the test results given in the Tables, it would be good to add Figures with selected test results. These Figures should make it possible to determine the tensile strength at smaller strains, e.g. 2%
- the data in Tables 8-10 looks like duplicate data shown in Table 7. What does Time (hour) 0.1 and 438,000 mean in Tables 8-10 where for the deformations mentioned earlier in the description to Table 7 one minute is given. The maximum creep strain 10.35 at 100% of UTS and 40% of creep load is too high in comparison to the rest of the values obtained for 50% and 60% of creep load. This requires additional comment. Wouldn't it be better to put Figures with selected test results instead of Tables 8-10?
Detailed comments and suggestions:
Page 3 Line 92: is: from 20°C to 14°C; should be: from 20°C to 76°C every 14°C
Page 4 Line 128-131: is: In the first (or transient) stage, it is called transition creep, and the initial deformation occurs and the strain decreases as the deformation increases. In the second (or steady state) stage, the displacement increases linearly with time, resulting in a constant strain.
Page 4 Line 128-131: should be: In the first (or transient) stage, it is called transition creep, and the initial deformation occurs and the strain increases non-linearly as the deformation increases. In the second (or steady state) stage, the displacement increases linearly with time, resulting in almost constant strain.
Page 5 Line 163: is: Tables 4.1 ; should be: ??
Page 6 Line 216: is: of 23 to 79°C; should be: see above
Page 7 Line 220: is: the maximum creep strain rate by; should be: the maximum creep strain by
Sincerely
Reviewer
Reviewer 3 Report
In this paper, tensile properties, creep strain, reduction factor, creep behavior and fatigue properties of LCP woven geotextiles were tested and analyzed. The research results are of interest to the readership of the journal. However, there are some issues in the current manuscript that I wish to raise here for the authors' consideration:
- The main creep curves of the geosynthetics should be constructed on the basis of the analysis of the creep deformation tests using the Stepped Isothermal Method.
- How to simulate the seismic load in the creep test? The authors should explain this explicitly in the revised manuscript.
Round 2
Reviewer 1 Report
Please improve your work carefully and take the tips seriously.
- Be sure to expand your literature review.
-Polymeric Synthetic Fabrics to Improve Stability of Ground Structure in Civil Engineering Circumstance; Han-Yong Jeon
- Vectran LCP Fibre; Rahul Rajkumar Gadkari & Miss Snehal Kasture
- Melt-Spun Fibers for Textile Applications; Rudolf Hufenus
e.t.c.
- Material data are available on materials. Compare with these results.
https://www.kuraray.eu/fileadmin/product_ranges/vectran/downloads/Kuraray_Vectran_composite2019_ToGo.pdf
https://www.kuraray.eu/fileadmin/product_ranges/vectran/downloads/2017_VECTRAN_Product_Brochure.pdf
- In the table below, the stress units are incorrectly stated as N / m2
